# Determination of Binding Affinity of Antibodies to HIV-1 Recombinant Envelope Glycoproteins, Pseudoviruses, Infectious Molecular Clones, and Cell-Expressed Trimeric gp160 Using Microscale Thermophoresis

**DOI:** 10.3390/cells13010033

**Published:** 2023-12-22

**Authors:** Shraddha Basu, Neelakshi Gohain, Jiae Kim, Hung V. Trinh, Misook Choe, M. Gordon Joyce, Mangala Rao

**Affiliations:** 1Henry M Jackson Foundation for the Advancement of Military Medicine, Bethesda, MD 20817, USA; sbasu@hivresearch.org (S.B.); ngohain@gmail.com (N.G.); jakim@hivresearch.org (J.K.); htrinh@hivresearch.org (H.V.T.); misook.choe@nih.gov (M.C.); gjoyce@eidresearch.org (M.G.J.); 2US Military HIV Research Program, Walter Reed Army Institute of Research, Silver Spring, MD 20910, USA; 3Emerging Infectious Disease Branch, Walter Reed Army Institute of Research, Silver Spring, MD 20910, USA

**Keywords:** HIV-1 epitopes, MST, binding affinity, recombinant HIV-1 proteins, pseudovirus, infectious molecular clone, cell-expressed trimeric protein, bNAb, mAb

## Abstract

Developing a preventative vaccine for HIV-1 has been a global priority. The elicitation of broadly neutralizing antibodies (bNAbs) against a broad range of HIV-1 envelopes (Envs) from various strains appears to be a critical requirement for an efficacious HIV-1 vaccine. To understand their ability to neutralize HIV-1, it is important to characterize the binding characteristics of bNAbs. Our work is the first to utilize microscale thermophoresis (MST), a rapid, economical, and flexible in-solution temperature gradient method to quantitatively determine the binding affinities of bNAbs and non-neutralizing monoclonal antibodies (mAbs) to HIV-1 recombinant envelope monomer and trimer proteins of different subtypes, pseudoviruses (PVs), infectious molecular clones (IMCs), and cells expressing the trimer. Our results demonstrate that the binding affinities were subtype-dependent. The bNAbs exhibited a higher affinity to IMCs compared to PVs and recombinant proteins. The bNAbs and mAbs bound with high affinity to native-like gp160 trimers expressed on the surface of CEM cells compared to soluble recombinant proteins. Interesting differences were seen with V2-specific mAbs. Although they recognize linear epitopes, one of the antibodies also bound to the Envs on PVs, IMCs, and a recombinant trimer protein, suggesting that the epitope was not occluded. The identification of epitopes on the envelope surface that can bind to high affinity mAbs could be useful for designing HIV-1 vaccines and for down-selecting vaccine candidates that can induce high affinity antibodies to the HIV-1 envelope in their native conformation.

## 1. Introduction

Since the beginning of the HIV-1 epidemic, almost 40.4 million people have died of HIV/AIDS, and 85.6 million people have been infected with this virus. According to UNAIDS, through the end of 2022, approximately 39 million (33.1–45.7 million) people were living with HIV around the world [1]. Therefore, the development of strategies, particularly vaccines, to prevent the spread of the virus remains a global public health priority. Despite several Phase 3 clinical trials and breakthroughs in HIV research, the quest for an efficacious HIV-1 vaccine continues. Even though it is known that antibodies play a crucial role as the first line of defense against virus entry, the characteristics of binding to the specific epitopes on HIV-1 envelope (Env) glycoproteins of different strains are not well understood. The characterization of monoclonal antibodies (mAbs) and/or broadly neutralizing antibodies (bNAbs) can also elucidate the vulnerable regions of the HIV-1 Env trimer. It will enable us to comprehend how mAbs neutralize the virus by binding to the epitopes on the Env glycoprotein, preventing viral entry and/or viral replication. This information can potentially aid in the design of an effective HIV-1 vaccine. 

The HIV-1 trimeric Env glycoprotein consists of three subunits of gp120 protein noncovalently anchored to gp41 protein [2]. Engagement of the primary CD4^+^ receptor and the CCR5 or CXCR4 chemokine coreceptors by trimeric gp120/gp41 is required for viral entry [3]. The binding of the Env trimer to CD4 exposes the CCR5 co-receptor binding site in the variable loop 3 (V3) of gp120 and the gp41 stalk, leading to the formation of a six-helix bundle conformation in gp41, facilitating viral and cell membrane fusion [4] and leading to virus entry [5,6,7]. 

People living with HIV (PLWH) produce antibodies that recognize the virus within months of infection, while heterologous neutralizing antibodies appear after 1 or more years [8,9]. However, due to the high rate of mutation and the latency of HIV-1, it is difficult to eliminate the virus in vivo as the virus is always a few steps ahead of the antibodies [10]. Many different bNAbs have been isolated from PLWH [11,12,13,14,15,16]. Recent strategies have utilized B-cell sorting and deep sequencing to isolate bNAbs as well as mAbs and their neutralizing abilities, and/or Fc effector functions have been demonstrated against a wide variety of HIV-1 strains [17,18,19,20,21]. These bNAbs and mAbs bind to different epitopes on the HIV-1 Env trimer and the glycans linked to them [22,23], including the CD4 binding site (CD4bs), V1V2, the glycan-dependent V3 region, the MPER region of gp41, the fusion peptide, and the interface region between gp41 and gp120 [24,25,26,27,28,29]. Multiple studies have shown that the passive transfer of bNAbs protects against infection or, in some cases, reduces viral loads or delays viral rebound in non-human primate and humanized mouse models [30,31,32,33,34]. However, to date, none of the candidate HIV-1 immunogens have succeeded in inducing bNAbs following vaccination in human clinical trials. The elicitation and presence of mucosal antibodies at the site of infection may be necessary to prevent virus entry, which may also require the elicitation of high affinity antibodies at the target site. Determinations of the affinity of HIV-1 bNAbs and other HIV-1 mAbs have generally been performed with recombinant HIV-1 Env proteins. Env proteins are highly diverse among HIV-1 subtypes and even within a subtype, mAbs show variable affinities. It is important to better understand their binding characteristics to design vaccines that would elicit high affinity antibodies. From a therapeutic standpoint, this information could aid in formulating a better cocktail of bNAbs and/or mAbs that would be more effective globally. HIV-1 vaccines have, so far, been unsuccessful at inducing high affinity bNAbs and in addressing whether they correlate with protection. However, for some parasitic, viral, and bacterial antigens, it has been demonstrated that the induction of high avidity antibodies following vaccination confers protection [35,36,37,38]. The induction of high avidity/affinity functional antibodies by HIV-1 vaccines could play a major role in protection against infection.

A robust and reliable determination of the affinity between a target molecule and its ligand is a critical step in understanding the magnitude of the binding interaction between an antibody and its antigen. There are several biochemical and biophysical techniques that can be utilized to determine the binding characteristics of antibodies to viral antigens. Two of the immobilization methods routinely used to determine the binding affinity of an antibody are surface plasmon resonance (SPR) [39,40] and Bio-layer interferometry—Octet (BLI) [41,42]. Although several different formats have been utilized to determine the binding affinities of bNAbs, a universal assay has not been developed to address the affinities determined by the various platforms. Thus, there are some gaps in understanding how binding correlates with function and how this is influenced by differences in HIV-1 Envs among the various subtypes and strains. Membrane forms of Envs can be examined by flow cytometry, cell ELISA, and VLP ELISA. The use of different platforms makes it difficult to compare mAb affinities to membrane or soluble forms of Env since the various forms of the Env cannot be analyzed using a single method. 

A more recently developed method for determining binding affinity is immobilization-free, in-solution-based Microscale Thermophoresis (MST). This technology utilizes the phenomenon of thermophoresis (defined as the movement of a molecule along a temperature gradient) by utilizing the thermophoretic movement of a target molecule which can be impacted upon binding with its ligand as there would be changes in mass and size [43,44,45]. Recently, MST has emerged as a biophysical technique for quantifying the binding affinities of biomolecules on a microliter scale [46,47,48]. MST can determine the equilibrium constant (K_D_) without the immobilization of either the target or ligand under almost native-like conditions with a very short measurement time. This method is also a purely optical method which requires samples to be loaded into capillaries and enables a contact-free environment which reduces the chances of contamination [46]. This technique requires small amounts of protein and can measure binding affinities at picomolar concentrations [49].

To date, MST has not been used to determine the binding affinities of HIV-1 bNAbs and mAbs with HIV-1 Env proteins. In the present study, we utilized MST to determine the binding affinities of mAbs and bNAbs with recombinant HIV-1 Env glycoproteins, cell-expressed HIV-1 trimers, pseudoviruses (PVs), and infectious molecular clones (IMCs). Any hindrance or lack of availability of specific epitopes for binding due to immobilization techniques are avoided with the MST technique. An investigation of these antibodies revealed differences in affinities between binding to recombinant monomer versus trimer Env proteins and PVs, IMCs, and Envs expressed on the cell surface. These differences could shed light on why certain antibodies bind strongly with a soluble protein in vitro but are not as effective with viruses, for which the conformation or display of the Env protein on the surface is important. It was previously reported that the induction of high affinity antibodies following vaccination is important for protection against influenza and malaria [50,51]. In view of this information, antibody affinity measurements could be important for evaluating HIV-1 vaccines and for the selection of high affinity bNAbs for therapeutic purposes. 

## 2. Materials and Methods

### 2.1. Reagents

HIV-1 subtype B Env proteins (SF162 gp140, and BaL gp120) were obtained from the NIH HIV Reagent Program, Division of AIDS, NIAID, NIH: Human Immunodeficiency Virus Type 1 SF162 gp140 Trimer Protein, Recombinant from HEK293T Cells, ARP-12026, contributed by Dr. Leo Stamatatos, and glycoprotein gp120 was obtained from Human Immunodeficiency Virus Type 1 (HIV-1) BaL, Recombinant from HEK293F Cells, HRP-20082, contributed by DAIDS, NIAID. SF162 gp120 was obtained from Dr. Venigalla B. Rao, The Catholic University of America. CRF01_AE A244 gp120 and A244 gp140 were obtained from Immune Technology Corp, New York, NY, USA. An A244 gp140 data sheet shows a strong band at 140 kD determined by Western blotting. A244 gp120 is full length, while A244 gp140 has 11 amino acids deleted at the N-terminal end. A BG505-SOSIP trimer Env (subtype A) was produced, purified, and provided by Dr. Gordon Joyce. BG505 SOSIP was verified as a trimer by gel-filtration and negative-stain EM. Stably transfected CHO-K1 cells expressing the acute subtype C (C6980V0C72) gp145 protein were produced and purified by ABL, Inc. (Rockville, MD, USA) under a contract. The preparation consisted mainly of trimers, with approximately 60% actual trimers exhibiting the fan blade motif, as determined by cryo-EM analyses, although we do not know if they are open or closed trimers [52]. The monoclonal antibodies (mAbs) DH827, CH58, and CH59 and the bNAbs PGT145, PGDM1400, PG9, 447-52D, 2G12, VRC01, 3BNC117, and 10E8 were produced and purified as described below. 

BaL and CM235 pseudoviruses (PVs) were prepared as previously described [53,54]. Briefly, 293T cells (5 × 10^6^) were transfected with 8 µg of an env expression plasmid and 24 μg of an env-deficient HIV-1 backbone vector (pSG3ΔEnv), using a FuGene 6 transfection reagent (Roche). The plasmids used for the PVs contained the pBR322 vector. The culture supernatants were harvested after 3 days, clarified by centrifugation at 311× *g* for 10 min, and passed through a 0.22 µm filter. The culture supernatant was subjected to ultracentrifugation at 100,000× *g* for 90 min at 4 °C, and the virus pellet was resuspended in 400 µL of PBS. The PVs were purified as previously described [55]. Briefly, exosomes and microsomes were removed using protein A/G beads pre-coated with anti-acetylcholinesterase, followed by the addition of an anti-CD45 antibody cocktail. PVs were stored in liquid nitrogen until use in aliquots containing 15% FBS that had been subjected to ultracentrifugation to remove bovine exosomes. 

A BaL Infectious Molecular Clone (IMC) plasmid was obtained through the NIH HIV Reagent Program, Division of AIDS, NIAID, NIH: Human Immunodeficiency Virus 1 (HIV-1) HXB3/BaL Infectious Molecular Clone (pWT/BaL), ARP-11414, contributed by Dr. Brian Cullen. The plasmid was further expanded (GeneScript, Piscataway, NJ, USA) and contained the WT-10 vector. A BaL IMC was generated by transfection of 293T cells with Fugene 6, and the culture supernatant containing the virus was harvested after 2 days. A CM235 IMC was generated by Dr. Sodsai Tovanabutra at MHRP and contained the pBC KS1 vector. IMCs were purified and stored as described above. Infectivity and p24 concentrations were determined before and after purification to ensure that infectivity was not lost during the purification procedure. The amount of viral p24 in the PSVs and IMCs was determined using the HIV-1 p24 Antigen Capture Assay kit (ABL, Rockville, MD, USA). PVs and IMCs were used in the MST assay based on the amount of p24. The following starting concentrations were used: BaL PV: 2 pg; BaL IMC: 83 pg; CM235 PV: 79 pg; CM235 IMC: 25 pg. One pg of virus equates to approximately 10^4^ virus particles. Previous work determined that there are 106.7 Env trimers per virion for BaL PV and 42.5 Env trimers per virion for CM235 PV [53]. Louder et al. determined that on an average, there are 8 Env trimers per virus particle [56]. Therefore, the total number of env trimer molecules loaded in the MST capillaries at the highest concentrations were as follows: 2 × 10^6^ (BaL PV), 6.7 × 10^6^ (BaL IMC), 3.3 × 10^7^ (CM235 PV), and 2 × 10^6^ (CM235 IMC). The serially diluted intact virus (1:1) and a fixed concentration of labeled mAbs were loaded in the MST premium capillaries to achieve the required binding curves. 

CEM-NKR 5001A cells expressing MN gp160 on their surface were produced and provided by Dr. Jeffrey Currier (unpublished data). The cells were grown in R10 (RPMI-1640, Quality Biological, Gaithersburg, MD, USA) media that contained 10% Fetal Bovine Serum (Gemini Bio Products, Sacramento, CA, USA), 2% L-glutamine, and 2% Penicillin/Streptomycin (Quality Biological, Gaithersburg, MD, USA). Protein-expressing cells were treated with 250 µg/mL hygromycin B (Sigma Aldrich, St. Louis, MO, USA), whereas parental cells were not treated with hygromycin B.

### 2.2. Plasmid Constructions, Antibody Production, Purification, and Characterization

Codon-optimized genes corresponding to the antibody heavy- and light-chain intermediates were synthesized with a 5′ secretion leader sequence and sub-cloned into the mammalian expression vector pVRC-8400. Antibody variable regions were synthesized and cloned (GenScript, Piscataway, NJ, USA) into CMVR expression vectors (NIH AIDS reagent program). Plasmids encoding heavy and light chains were synthesized, transformed into DH5α cells (GeneScript, Piscataway, NJ, USA), and co-transfected into Expi293F cells (ThermoFisher, Waltham, MA, USA) with equal amounts of Ig heavy (VH) and light (VL) chain genes using poly-ethylenimine (PEI) or with 0.4 mg of heavy- and light-chain genes using ExpiFectamine™ (Life Technologies, Carlsbad, CA, USA), following the manufacturer’s protocol. After five days, antibodies were purified from clarified culture supernatants using a Protein A agarose column (ThermoFisher Scientific, Waltham, MA, USA). The antibodies were eluted with PBS adjusted to pH 2.8, and the eluate was collected in tubes containing Tris-HCl buffer pH 8.0. The antibodies were buffer-exchanged in PBS and quantified by measuring the absorbance (A) at A280 nm. The purity and integrity of the mAbs were assessed using SDS-PAGE and Coomassie Blue staining under both reducing and non-reducing conditions. The pooled eluates were concentrated and subjected to the final step of purification using gel filtration chromatography on a Superdex 200 16/60 column (GE Healthcare, Piscataway, NJ, USA). The fractions collected were characterized for their purity using SDS-PAGE, and the pure fractions were pooled, concentrated, quantified at A280 nm, aliquoted, and stored at −80 °C until further use. 

### 2.3. Microscale Thermophoresis (MST)

The binding affinity of the purified mAbs was determined by MST using the Monolith NT.115 Pico (Nano Temper Technologies, Cambridge, MA, USA). Antibodies used for MST were labeled using the NanoTemper protein-labeling red maleimide kit (NanoTemper Technologies, Cambridge, MA, USA), which labels the sulfhydryl -SH group on the cysteine residues and was used as described in the manufacturer’s provided protocol. The concentration of the purified, labeled antibodies was determined by measuring the absorbance at 280 nm and 650 nm using a Nanodrop (ThermoFisher, Waltham, MA, USA). Pretests of the labeled antibodies were performed on the MST Monolith NT.115 Pico to check for the quality of the fluorescent-labeled antibody. Antibodies were diluted to different concentrations (100 nM, 50 nM, 25 nM, 12.5 nM, and 6.5 nM) in PBS buffer containing 0.01% Tween 20 until the desired concentration at which the fluorescence counts were ~4000–8000 was found. The quality check and the MST trace for a fluorescently labeled mAb, DH827, are shown in Appendix A. Having determined the optimum concentration of labeled DH827, this antibody concentration was utilized for a binding check with acute subtype C gp145 (C6980V0C72) protein (Appendix A). If binding of the labeled DH827 mAb with the gp145 protein was detected and the binding interaction between the fluorescently labeled DH827 mAb-gp145 complex resulted in a smooth MST dose–response curve (Appendix A), then the experiment was conducted to determine the binding affinity of the mAb. A quality check was conducted and binding curves were produced for each pair of labeled mAb and ligand (protein, viruses, cells) before proceeding to determine the binding affinity. Measurements were carried out immediately or after an incubation period of 30 min or 60 min at room temperature or after 30 min or 60 min at 37 °C to determine the optimum time and temperature.

The MST power and LED excitation power were both set at 60%. Measurements were carried out in PBS buffer containing 0.01% Tween 20. For the binding affinity assay, 16 premium capillaries were loaded with a constant concentration of the fluorescently labeled mAb along with serially diluted concentrations of either the HIV-1 Env protein, cells expressing gp160, PVs, or IMCs (antigen). PVs, IMCs, and cells were not lysed and used as viral particles or intact cells. Intact PVs and IMCs were used at starting concentrations of 2 × 10^6^ BaL PV, 6.7 × 10^6^ BaL IMC, 3.3 × 10^7^ CM235 PV, and 2 × 10^6^ CM235 IMC and were each individually serially diluted in PBS buffer containing 0.01% Tween 20 (1:1). CEM-NKR 5001A cells expressing MN gp160 on their surface or control cells not expressing the Env protein were each used at a starting concentration of 5 × 10^6^/mL and serially diluted 1:1 with buffer, as mentioned above. The labeled mAbs were used at a fixed, pre-determined concentration, and measurements were carried out after an incubation period of 30 min at RT to achieve the required binding curves. Compared to the unbound labeled monoclonal antibody, the labeled mAb–Env complexes move at a slower rate through the temperature gradient (Appendix A). The signal was analyzed after the start of the IR laser. Based on the movement of the fluorescently labeled mAb, the binding affinity of the mAb was determined using MO. affinity analysis software (V2.2.7) provided by NanoTemper. 

### 2.4. Statistical Analysis

Each experiment with one set of labeled mAb and Env protein, IMC, PV, or cell expressing gp160 was repeated 2–3 times. Mean and standard deviation (SD) values were determined for each set. To determine significant differences between the sets, unpaired *t*-tests were performed using GraphPad Prism software (version 10.0.1). To determine if there was a correlation between the binding affinities determined with MST (K_D_) and reported neutralization (IC_50_), the two values for each mAb were plotted, and the Pearson correlation was determined using GraphPad Prism software (version 10.0.1). 

## 3. Results

### 3.1. Structural Model of HIV-1 Envelope Epitopes Recognized by Broadly Neutralizing and Non-Neutralizing Monoclonal Antibodies

To visualize the epitopes of the HIV-1 Env targeted by bNAbs and non-neutralizing mAbs, a structural model was generated based on the PDB code 5FYJ using the PyMOL program (Figure 1). The surface representation of the HIV-1 Env was derived from the crystal structure of BG505.664 (PDB code: 5FYJ). The major regions of the HIV Env targeted by the mAbs used in our experiments are highlighted in different colors and represent the following: V1V2 epitopes (purple) recognized by PG9, PGT145, PGDM1400, DH827, CH58, and CH59; a glycan patch (magenta) on the V3 loop recognized by 447-52D and 2G12; a CD4 binding site (brown and yellow) recognized by VRC01 and 3BNC117; and the MPER region (dark green) recognized by 10E8.

### 3.2. Determination of the Effects of Time and Temperature on the Binding Affinities of Broadly Neutralizing Antibodies (bNAbs) with Recombinant HIV-1 Envelope Proteins by Microscale Thermophoresis (MST)

MST technology utilizes a fluorochrome-labeled antibody and determines binding affinity by detecting variations in the fluorescence signal as a result of an infrared laser-induced gradient temperature change between the labeled antibody and the Env protein. The bNAbs were labeled with a cysteine reactive red maleimide kit from NanoTemper as directed by the manufacturer. We utilized subtype B (BaL and SF162) gp120 and gp140 proteins and the subtype CRF01_AE (A244) gp120 protein to determine if the binding affinities of the CD4bs-specific bNAb VRC01 and the glycan antibody 2G12 were influenced by the incubation time or temperature. The time of incubation (30 min or 60 min) did not change the binding affinity of VRC01 to BaL gp120, A244 gp120, and SF162 gp140 (Figure 2A,B,E,F,I,J). Similarly, the time of incubation did not influence the binding affinity of 2G12 to BaL gp120, A244 gp120, and SF162 gp140 (Figure 2C,D,G,H,K,L). 

We then determined the effect of temperature on the binding affinities of VRC01 and 2G12 using the same proteins as above (Figure 3). Incubation for 30 min at RT or at 37 °C resulted in similar binding affinities for VRC01 (Figure 3A,B,E,F,I,J) and 2G12 (Figure 3C,D,G,H,K,L) at the two temperatures with the three proteins tested. The curves are graphed as Fnorm versus the concentration of the Env protein. The Fnorm value is associated with the fluorescence signal of the labeled antibody. The curves reflect the data from an unbound state of the labeled antibody to the bound state of the labeled antibody-Env complex. In all cases, the unbound state of the labeled antibody has a higher fluorescence signal (Fnorm) than in the bound state of the antibody-Env complex. However, in some cases, as seen with VRC01 and the SF162 gp140 protein (Figure 3I), the unbound state has a lower Fnorm, and the Fnorm signal increases in the bound state, which is the opposite of the other curves. The binding affinity is determined by the maximum and minimum Fnorm values determined from the unbound and bound states and therefore, the opposite curve with VRC01 does not affect the determination of the binding affinity. These results indicated that the binding affinity was independent of the incubation time and temperature. All further experiments were carried out at RT for 30 min. 

### 3.3. Determination of the Binding Affinities of bNAbs and mAbs to Recombinant Monomer and Trimer HIV-1 Env Proteins by MST

Having determined that the binding affinities were not influenced by the time or temperature of incubation, we utilized the mAbs depicted in Figure 1 to determine the binding affinities to the monomeric or trimeric Env proteins of various HIV-1 subtypes (CRF01_AE, A, B, and acute C) (Appendix A, Table 1). We utilized bNAbs targeting the various epitopes on the HIV-1 Env as well as non-neutralizing mAbs DH827 isolated from the RV305 Phase 2 clinical trial and CH58 and CH59 [40] isolated from the RV144 Phase 3 clinical trial. Representative MST traces with the respective K_D_ values are shown in Appendix A. The V1V2 targeting bNAb PGT145 bound to the monomeric Env proteins A244 gp120, BaL gp120, and SF162 gp120 with very tight affinity (less than 20 nM) but exhibited weaker binding affinity to the trimeric A244 gp140 and SF162 gp140 proteins (Appendix A and Table 1). PGT145 demonstrated the strongest binding to BG505 SOSIP, while no binding was observed with the acute subtype C (C6980V0C72) gp145 protein, which may be due to the closed conformation or folding of the SOSIP trimer (Appendix A and Table 1). In contrast, mAb DH827 showed the strongest binding to the C6980V0C72 gp145 protein, with no binding to BaL gp120 and BG505 proteins (Appendix A and Table 1). The V3-specific mAb 447-52D did not exhibit any binding to gp120 A244 but bound to the other monomer and trimeric proteins tested (Appendix A and Table 1). The VRC01 mAb bound to all the proteins tested (Appendix A and Table 1). To demonstrate the specificity of binding of HIV-1 mAbs to HIV-1 Env proteins, we utilized the mAb Synagis, directed against respiratory syncytia virus (RSV), as a negative control mAb. No binding was observed between the labeled Synagis mAb and the HIV-1 Env proteins A244 gp120, A244 gp140, and SF162 gp120 (Appendix A). 

To visualize and compare the binding affinities of the mAbs determined by MST to the soluble recombinant proteins, the binding affinities were graphed as a heat map (Figure 4A). The darker the color, the higher the binding affinity. A lack of binding between the labeled mAbs and the Env proteins is represented by white color. The quaternary V1V2-specific bNAb PGDM1400, as previously demonstrated [57], did not exhibit binding to subtype B and CRF01_AE gp120 and 140 proteins (Figure 4A, Table 1) but bound with high affinity to both subtypes, C and A trimeric proteins (Figure 4A and Table 1). The V1V2 region-specific bNAb PG9 exhibited differential binding characteristics compared to PGT145 and PGDM1400. Although PG9 preferentially binds trimers [58], in contrast to PGDM1400, it bound the HIV-1 Env monomers A244 and S162 gp120 with low affinity. PG9 bound with very high affinities to the A244, SF162, and acute C trimeric proteins with the exception of BG505 SOSIP (Figure 4A and Table 1). 

The three non-neutralizing mAbs (DH827, CH58, and CH59) that recognize linear epitopes on V2 did not bind to the BaL gp120, SF162 gp140, and BG505 SOSIP proteins (Figure 4A and Table 1). As expected, all three mAbs bound to A244 gp120 with low nanomolar affinity. In addition, DH827 exhibited weaker affinity to SF162 gp120 compared to A244 gp120. Surprisingly, DH827 bound to the trimeric protein acute C gp145 with high affinity and to A244 gp140 with relatively low affinity. CH59 did not bind any of the trimers examined. In contrast, CH58 bound with strong affinity to acute C gp145 protein but did not bind to the other trimers tested (Figure 4A, Table 1). 

We examined the binding interaction of the V3 loop targeting the bNAb 447-52D with recombinant HIV-1 Env proteins (Figure 4A, Table 1). We found that 447-52D bound to SF162 gp120 with high affinity (Figure 4A, Table 1). However, the binding affinity with BaL gp120, another subtype B protein, was much lower, and no binding was observed with A244 gp120 (Figure 4A, Table 1). Although 447-52D bound to SF162 gp120 efficiently, no binding was obtained with the corresponding trimer. Additionally, 447-52D bound to three out of the four HIV-1 trimers tested, exhibiting the strongest binding affinity to BG505 SOSIP (K_D_ = 0.25 ± 0.03 nM) (Figure 4A, Table 1).

2G12 is a bNAb that recognizes the glycans on the surface of the HIV-1 Env [59]. 2G12 bound to all the monomers and trimers tested with high affinity. The K_D_ ranged from 0.12 nM (C6980V0C72 gp145) to 22.35 nM (A244 gp140) (Figure 4A, Table 1). This may be due to the ability of 2G12 to recognize the conserved epitope on gp120 that is composed of oligomannose-type glycans. Furthermore, the nearby glycans might influence the recognition process. 

We next examined the CD4bs bNAbs VRC01 and 3BNC117. VRC01 bound to all the monomer and trimer proteins tested, some with slightly higher affinity (Figure 4A and Table 1). VRC01 bound to the A244 gp120 monomer with a similar binding affinity compared to its equivalent trimer, while VRC01 bound to the SF162 gp120 protein with a slightly stronger affinity compared to its equivalent trimer (Figure 4A and Table 1). Comparable high binding affinities were observed with the BG505 SOSIP and C6980V0C72 gp145 proteins (Figure 4A and Table 1).

In contrast to VRC01, 3BNC117 showed differential binding affinities to HIV-1 monomers and trimers. The binding affinities (K_D_) of 3BNC117 to A244, BaL, and SF162 gp120 and A244 gp140 ranged from 35 nM to 60 nM (Figure 4A, Table 1). As was observed with VRC01, 3BNC117 displayed the strongest binding affinities to C6980V0C72 gp145 (K_D_ = 3.36 ± 1.95 nM) and BG505 SOSIP trimers (K_D_ = 7 ± 1 nM) (Figure 4A, Table 1). 

10E8 is a bNAb specific for the conserved membrane proximal external region (MPER) of the HIV-1 Env protein [60]. Monomeric proteins do not contain the gp41 region; thus, 10E8 did not bind to any of the monomeric proteins tested (Figure 4A, Table 1). The 10E8 bNAb did not bind to BG505 SOSIP but bound to the other trimeric proteins tested. An extremely strong binding affinity was observed with the C6980V0C72 gp145 protein (K_D_ = 0.59 ± 0.03 nM) (Table 1). In contrast, 10E8 bound with much lower affinities to the A244 gp140 and SF162 gp140 proteins (Figure 4A and Table 1). 

### 3.4. Differential Binding Affinities of Monoclonal Antibodies (mAbs) to Pseudoviruses (PVs) and Infectious Molecular Clones (IMCs)

An examination of mAbs and their binding affinities to HIV-1 PVs and IMCs will yield important information regarding their ability to inhibit the virus from entering the cell. However, these interactions are not possible to examine with SPR because SPR requires ligand immobilization. Alternatively, MST is an immobilization-free method and binding is detected in-solution. Therefore, we utilized MST to determine the binding affinities of mAbs and bNAbs with BaL (Appendix A) and CM235 PVs and IMCs (Appendix A), and the respective binding curves with K_D_ values are shown. The binding affinities for the mAbs and the viruses were graphed as a heat map on the same scale as the heat map for the recombinant HIV-1 Env proteins (Figure 4B). PGT145 exhibited binding to BaL PVs and IMCs with K_D_ values of 34 nM and 22 nM, respectively (Appendix A and Figure 4B). Although PGT145 bound to the CM235 PV (K_D_ = 15 nM), it did not bind to the CM235 IMC. (Appendix A and Figure 4B). Similarly, VRC01 demonstrated a 34-fold significantly higher binding affinity (*p* = 0.001) to the BaL IMC (K_D_ = 1.24 ± 0.16 nM) than to the BaL PV (K_D_ = 41.75 ± 6.47 nM) (Appendix A and Figure 4B). We observed differential binding affinities for the mAbs tested with the CM235 PV and IMC compared to the A244 gp140 protein. DH827, PGT145, and 447-52D mAbs exhibited fold higher binding affinities to the CM235 PV compared to the CM235 IMC (Appendix A and Figure 4B). PGT145 showed comparable binding affinities to the CM235 PV and A244 gp120 protein (Appendix A and Figure 4B, and Table 1). 447-52D showed a 4-fold lower binding affinity to the PV compared to the A244 gp140 protein (Appendix A and Figure 4B and Table 1). PGT145 and 447-52D exhibited no binding to the CM235 IMC. VRC01 and 3BNC117 bound to both the CM235 PV and IMC (Appendix A and Figure 4B). The binding affinities of VRC01 to the CM235 PV and IMC were much weaker compared to the protein (Appendix A and Figure 4B and Table 1). 3BNC117 showed comparable binding affinities to IMC and PV (Appendix A and Figure 4B) and to the gp120 A244 protein (Figure 4A, Table 1). The bNAb 10E8 did not bind to the CM235 PV but bound to the IMC (Appendix A and Figure 4B) with a more than 5-fold higher affinity (*p* = 0.002) compared to the A244 gp140 protein (Figure 4A and Table 1).

### 3.5. Distinctive Binding Affinities of mAbs to Cell-Expressed MN gp160 

To determine if the binding affinities of the mAbs to cell-expressed HIV-1 Env proteins differed from the binding affinity to recombinant soluble proteins, MST was utilized to evaluate the binding affinities of six mAbs that recognized the V1V2, V2, V3, or MPER region to subtype B MN gp160 expressed on the surface of CEM cells (Figure 5). The corresponding MN trimeric protein was not available, so the data were compared to the SF162 (subtype B) gp140 recombinant protein (Figure 4A, Table 1). The specificity of mAb binding was confirmed by utilizing the parental cell line (CEM.NKR.CCR5) which did not express gp160 on the cell surface. No binding was observed with DH827 and 447-52D mAbs to the parental cell line (Appendix A). All six of the mAbs bound to cell surface-expressed gp160 with much higher affinities compared to the recombinant protein except for PG9, which showed a higher affinity with the recombinant protein (Figure 4A). In general, the binding affinities for the six mAbs with the cell-expressed MN gp160 were quite high and ranged from less than 1 to 40 nM (Figure 5).

### 3.6. Correlation between Neutralization and Binding Affinities of Monoclonal Antibodies Determined by MST

The mAbs examined in this study exhibit neutralizing (bNAbs) or Fc effector functions (non-neutralizing mAbs). Therefore, we determined if there was a correlation between the neutralization capabilities of these mAbs and the binding affinities determined by MST. We obtained the neutralization data using the CATNAP database for all the bNAbs examined in our study with the respective strains of viruses for the Env proteins used for the binding affinity measurements carried out by MST. The neutralization data were plotted as a heat map (Figure 6). Potent bNAbs have IC_50_ values below 3 µg/mL. Antibody/virus combinations that did not have neutralization data available are depicted in gray (Figure 6). The neutralization IC_50_ values for the mAbs were plotted against the binding affinities for the monomeric and trimeric Env proteins determined by MST (Figure 7A). Pearson’s correlation analysis showed that there was negative correlation which was not significant (r = −0.1; *p* = 0.63). We also plotted the neutralization IC_50_ values for the mAbs against the binding affinities for the viruses and the cell-expressed gp160 determined by MST (Figure 7B). The resulting Pearson correlation was positive (r = 0.3; *p* = 0.28), which was also not significant. These data indicate that there is no significant correlation between neutralization potency and binding affinities determined by MST across various bNAbs and virus strains. 

## 4. Discussion

Several studies have extensively characterized HIV-1 bNAbs and mAbs and their binding affinities utilizing soluble recombinant HIV-1 Env proteins [39,40,41,61,62,63]. However, when Env trimers are present in potentially more native conformations or on a virus particle, binding to antibodies could differ significantly. In the present study, we utilized MST, an emerging and sensitive immobilization-free in-solution-based method to quantitatively analyze the binding affinities of mAbs and bNAbs to HIV-1 Env proteins in different formats: soluble, on PVs and IMCs, and on the surface of cells. To the best of our knowledge, such a study has not been previously carried out. MST is rapid and possesses high-throughput potential compared to other methods such as isothermal titration calorimetry (ITC). Although ITC requires no labeling or immobilization, it requires relatively large amounts of the sample [64], in addition to having buffer limitations [65]. 

The observed binding affinities for the bNAbs were independent of two critical conditions that could affect the binding interactions. Neither the incubation time (30–60 min) nor the incubation temperature (RT and 37 °C) at which the experiments were conducted influenced the binding affinities of the bNAbs tested. A comprehensive examination of the binding affinities of HIV-1 soluble recombinant monomeric and trimeric Env proteins from different subtypes and strains with various epitope-specific mAbs and or bNAbs revealed that not all mAbs bound to both monomeric and trimeric proteins (Figure 4A). Differences in binding within the same subtype were observed with PG9 and DH827, with binding seen to SF162 gp120 but not to BaL gp120. In general, the affinities for the mAbs were much stronger for the trimers compared to the monomeric proteins.

There are two categories of V2-specific mAbs, DH827, CH58, and CH59, that recognized linear V2 epitopes and demonstrated stronger binding to gp120 proteins and exhibited either low affinity or no binding to trimeric proteins. This could be due to the linear epitopes being more easily accessible in the monomeric gp120 protein, while they may be obscured or not easily accessible in the trimer. DH827 has been reported to recognize an extended helical form of the V2 peptide [21]. CH58 and CH59 are gp120-strain-specific and bind only to the A244 protein. The monomer and strain specificity were maintained by CH59. CH58 bound with high affinity to gp145 (C) but not to the other gp140 proteins tested. DH827 also bound with very high affinity to gp145 (C) as well as to the Env (CRF01_AE) on the PV and IMC. This is probably because DH827 recognizes noncontinuous amino acid residues [21]. None of the three mAbs demonstrated pseudovirus neutralizing activity, with all viruses examined except for the Tier 1 92TH023 virus, but showed strong Fc effector function such as ADCC against tier 2 HIV-1 infectious molecular clone infected target cells [40] and ADCP responses. Our results with MST were generally consistent with previous reports using other assays such as ELISA, SPR [66], or beads-based assays [67] that showed that the V2-specific mAbs CH58 and CH59 were recognized exclusively by the V2 linear epitope specific subtype CRF01_AE of the A244 monomeric HIV-1 Env glycoprotein and not with the equivalent trimer. 

The conformational mAbs PGT145, PGDM1400, and PG9 showed varying degrees of binding to the different Env proteins tested but in general, the binding affinity was several folds higher with gp140 proteins compared to gp120 proteins. PGT145 also demonstrated binding to subtype B PVs and IMCs as well as to Subtype CRF01_AE PVs but not to IMCs. There are several published reports of these three mAbs demonstrating good pseudovirus and IMC neutralization.

As expected, the CD4bs-specific bNAbs VRC01 and 3BNC117, and the glycan dependent V3 epitope-specific bNAb 2G12, bound to all monomeric and trimeric recombinant proteins tested with variable binding affinities and, in some cases, demonstrated tighter binding with trimeric proteins. As expected, the MPER-specific bNAb 10E8 did not bind to the gp120 proteins due to the lack of the MPER region. Although it was previously shown that 10E8 neutralizes BG505 [68], we did not observe the binding of 10E8 to the BG505 SOSIP trimer. This is because the BG505 SOSIP protein we utilized for the MST studies did not have the complete 10E8 epitope. As with the SOSIP design, the C-terminus of the Env was truncated at residue 664, while the 10E8 epitope extends to residue 683 with significant contacts in the region from residue 668 onwards, which was not present in the BG505 SOSIP protein used. 

The V3 region bNAb 447–52D recognizes an epitope at the apex of the V3 loop [69]. The minimal epitope of 447-52D is GPGR [70]. 447-52D exhibited differential binding to the A244 gp140 and gp120 proteins. No binding was observed with the CM235 IMC as well. It is possible that the angle at which the antibody binds to the V3 apex epitope is important and that may not be conducive in A244 gp120 or in CM235 IMCs compared to A244 gp140. 

The binding affinities for several bNAbs were reported using surface plasmon resonance (SPR), octet, or ITC. Similarities as well as differences in the K_D_ values determined by MST and other methods are depicted in Appendix A. The table also highlights the differences in K_D_ values determined for bNAbs and mAbs by a single method, for example, SPR. It appears that MST and other methods showed comparable binding affinities in the low nanomolar range (Appendix A). However, it is difficult to make comparisons across the different platforms because of variability in the expression system and protein sequences or modifications of the protein. In the case of A244, certain publications have utilized the intact A244 protein, whereas others have used the protein with 11 amino acids deleted. Every method has some advantages and some drawbacks. In MST, the antibody is fluorescently labeled to detect changes in thermophoretic movements, which allows for binding affinity determination. If the fluorophore is present in the region of the Fab molecule that binds to the Env, the dye could either interfere with the binding or partially inhibit access to the Fab region by the epitopes on the Env, thus decreasing binding affinity. Variability within a method may also arise, as seen with SPR. The protein utilized in SPR has often been produced with a His-tag or an Avi tag. Thus, the Env protein can be captured by immobilizing an anti-His mAb, Biotin, or anti-human Fc antibody [39,62,71]. These differences could result in complete or partial accessibility of the epitopes, which could result in variations in the binding affinities determined. Although the binding of bNAbs and mAbs to Envs expressed on the surface of viruses or on cells has been demonstrated by flow cytometry, to the best of our knowledge, the binding affinities for the same have not been evaluated. 

One of the key advantages of MST over SPR or Octet is the ability to perform binding experiments in an enclosed and disposable capillary system, which allows for the capability to utilize other potentially infectious agents such as intact viruses or cells, cell lysates, and liposomes. This also eliminates the contamination of the equipment, the clogging of tubing, and safety concerns when working with potentially infectious material. MST allowed us to determine if there were differences in the binding affinities of the mAbs to gp160 Env trimers in their native conformation on the surfaces of PVs, IMCs, and cells compared to recombinant soluble proteins. We believe that this is the first time such a study has been conducted. PVs and IMCs belonging to two different HIV-1 subtypes, B (BaL) and CRF01_AE (CM235), which were propagated by the transfection of HEK293T cells, were utilized in this study. Overall, the PVs and IMCs showed lower K_D_ values compared to recombinant soluble proteins, and the IMCs showed stronger affinities to the mAbs tested compared to PVs. A broader examination was performed with the CM235 viruses which revealed that there was differential binding between the two types of viruses for certain bNAbs. PGT145 and 447-52D only bound to the PV and not to the IMC and, in contrast, 10E8 only bound to the IMC but not to the PV. The observed differences could arise from the fact that the PVs and IMCs were produced from different plasmids. The PVs were produced by co-transfecting a separate Env plasmid and a backbone plasmid containing the rest of the HIV genome, while the plasmids for IMCs contained the entire HIV genome, including the integrated Env gene of interest, which may have resulted in the conformations of the gp41 ectodomains (gp120 and gp41 through the MPER) being different. The amount of Env genomic DNA may also vary in the transfection for the PVs and IMCs, and this could in turn affect the overall number of Env trimers present on the surface of the virus. It was previously reported that the amount of virion associated with Env and the relative percentage of cleaved Env differed significantly between the PVs and IMCs, with frequently lower levels of incorporation of the Env and lower levels of cleaved gp120 Env on PVs. The variation in the presence of the total Env in the cleaved gp120 form could be as much as 90% in the IMCs, while in the PVs, it was much lower, with only 24% of the Env in the cleaved form [72]. It would be difficult to control this variable as well as to determine the proportion of trimers to monomers and the trimers in open and closed conformations. Nonetheless, the results show that the binding affinities of Envs on viruses and cells can be easily measured even with a very small number of viral particles. Future studies will aim to account for these variables as well as to extend the studies to Tier 2 and Tier 3 viruses and Env proteins.

The percent identity of the V2 loops and of the HIV-1 gp120 envelopes on the CM235 viruses and A244 protein have 95% sequence identity and, more specifically, the gp120 sequences for the two gp120 Envs have 94.2% sequence identity [73]. The CM235 and A244 V2 loops vary by two amino acids at positions 188 and 189 (HXB2 numbering), but their antibody-binding mid-regions are identical [74]. Thus, we can make some general comparisons with regards to the binding affinity of the CM235 PV and IMCs with the A244 gp140 protein. In some cases, the binding affinities of the bNAbs and mAbs tested against the viruses and soluble recombinant proteins are comparable. A good example is 3BNC117, for which the K_D_ values were 35 nM, 53 nM, 32 nM, and 26 nM for A244 gp120, A244 gp140, the CM235 PV and the CM235 IMC, respectively. In the cases of DH827 and 10E8, the binding affinities were higher with the viruses compared to the A244 gp140 protein, while the opposite was true for VRC01, thus underscoring the importance of determining the binding affinities of HIV-1 envelope-targeting mAbs with HIV-1 virus particles. 

Finally, we examined the binding affinities of bNAbs and the mAb DH827 to subtype B MN gp160 expressed on the surface of CEM cells which would have close to the native-like conformation of the HIV-1 envelope trimers. MST demonstrated its advantage as we were able to utilize intact cells in the capillary system to measure the binding affinity, which is not possible with other techniques. All the antibodies we examined bound to the cell-expressed Env trimers. No binding was observed with the parental cell line (negative control), thus demonstrating the specificity of binding. The amino acid sequence identity and similarity between MN gp160 and SF162 gp160 are 84% and 90%, respectively. Therefore, we were able to make general comparisons with the binding affinities determined for the cell-expressed MN gp160 and soluble recombinant SF162 gp140. Except for PG9, all the antibodies demonstrated very tight binding and had affinities which were even up to over 200-fold higher with cell surface-expressed MN gp160 compared to SF162 gp140. PGDM1400, DH827, and 447-52D did not bind to SF162 gp140 but exhibited very high binding affinities to cell surface-expressed MN gp160. These differences may be due to the accessibility of the epitopes or differences in glycosylation patterns, which could affect the conformation of the protein since SF162 gp140 was produced in 293T cells.

The heat maps that we generated with the binding affinity data for the mAbs demonstrate that the binding affinities for the recombinant proteins may not reflect the binding affinities for the viral envelopes present on either a PSV, IMC, or a cell surface. Furthermore, Pearson correlation analyses indicated the absence of a correlation between the binding affinity data generated using MST and the published neutralization data obtained from the CATNAP database. It is important to note that we performed the analysis with all the virus strains and subtypes together, which provided more of a global picture of the data. Previous reports showed a correlation between mAbs binding to BG505 SOSIP trimers and the BG505 virus. However, this was not with binding affinity data (K_D_) but with the EC_50_ for binding concentration [75]. We previously showed that neither binding nor the number of envelope trimers present on the virion were an indicator of virus neutralization by the mAb 4E10 [53].

Although we were not able to utilize the identical proteins in monomeric and trimeric forms, as well as IMCs, PVs, and cell surface-expressed proteins, we were able to compare the Env belonging to the same subtype in all formats tested. In summary, our findings demonstrate that MST can serve as a favorable method for investigating the binding antibodies elicited by either natural infection or HIV-1 vaccines and allow for the selection of vaccines that induce high affinity antibodies, thus aiding in HIV vaccine design and development. MST is a rapid, economical, and flexible alternative to conventional biophysical tools like SPR or octet. 

Future studies will include evaluations of immune sera or plasma samples for the presence of high affinity antibodies using the cost-effective and rapid MST assay. We envision that MST will complement the findings achieved through other methods, such as SPR or octet. Moreover, additional studies with MST could enhance our understanding of bNAb and mAb binding characteristics with the use of primary HIV-1 or the panel of PVs used in the TZM-Bl assay to determine if there is any correlation between HIV-1 neutralization and avidity/affinity. This gain in knowledge with binding studies of an Env in a native conformation, as measured by MST, could significantly improve and impact HIV-1 vaccine design.

## Figures and Tables

**Figure 1 cells-13-00033-f001:**
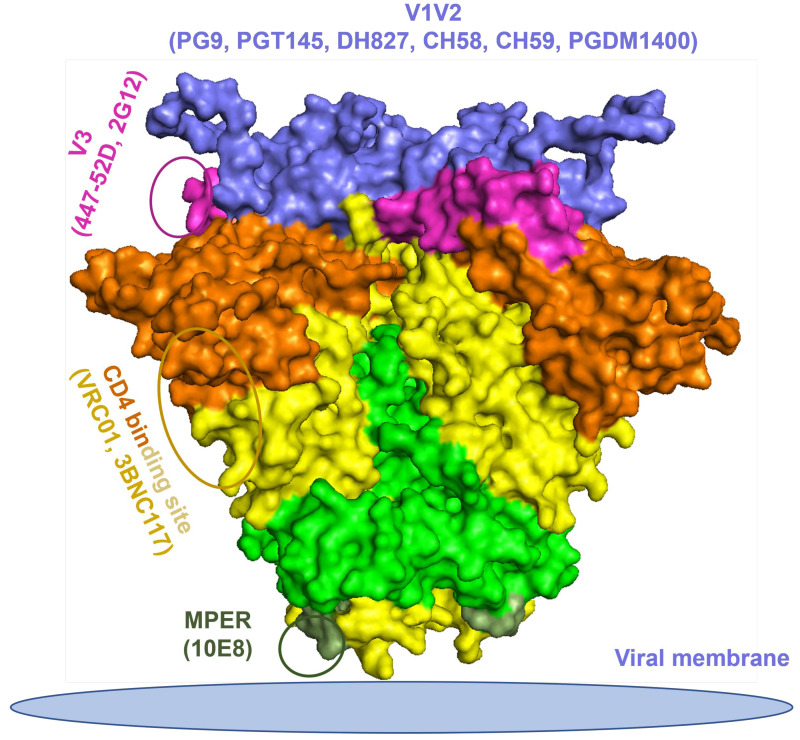
HIV-1 envelope glycoprotein spike trimer epitopes that bind various broadly neutralizing and non-neutralizing monoclonal antibodies (bNAbs and mAbs). The surface representation of the Env is derived from the crystal structure of BG505 SOSIP.664 (PDB code 5FYJ). The HIV-1 Env epitopes are highlighted in different colors: CD4bs (brown and yellow), V1/V2 (purple), V3/Asn332 glycan patch (magenta), MPER (dark green). Other unspecified regions are yellow for the gp120 moiety and light green for the gp41 moiety. The structural figure was generated with the program PyMOL, using the PDB code 5FYJ.

**Figure 2 cells-13-00033-f002:**
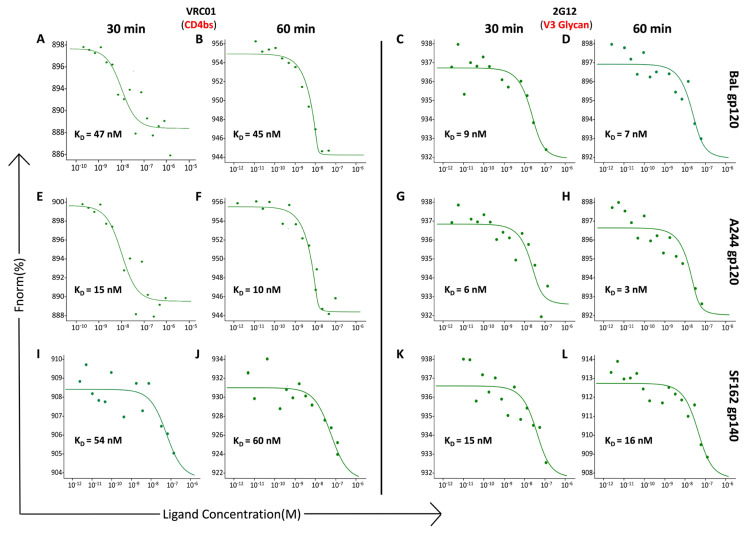
Impact of incubation time on the binding affinity of monoclonal antibodies (mAbs) to HIV-1 envelope proteins. Following a binding check, a constant concentration of labeled mAbs (VRC01: 12.5 nM; 2G12: 15 nM) in DPBS buffer without Ca^2+^ and Mg^2+^, pH 7.2, and containing 0.01% Tween-20 was used for determining the binding affinity. Sixteen premium capillaries were loaded with VRC01 or 2G12 and serially diluted (1:1) HIV-1 Env proteins, BaL gp120, A244 gp120, and SF162 gp140 (100 nM, 150 nM, and 75 nM, respectively), in a final volume of 10 µL. A Microscale Thermophoresis (MST) signal was then acquired using a NanoTemper Technologies Monolith NT.115 Pico instrument at an excitation power of 60% and an MST power of 60%. The signal was analyzed after the start of the IR laser, and the obtained data were fitted using MO. affinity analysis software provided by NanoTemper. The labeled mAbs and the serially diluted Env proteins were each incubated for 30 min or 60 min at RT and then loaded into the capillaries for affinity measurement. MST traces and binding affinity data are as shown: (**A**,**B**,**E**,**F**,**I**,**J**) VRC01 and (**C**,**D**,**G**,**H**,**K**,**L**) 2G12 with BaL gp120 (**A**–**D**); A244 gp120 (**E**–**H**); and SF162 gp140 (**I**–**L**), respectively. Relatively similar binding affinities were obtained at the incubation times tested. The graphs are presented as Fnorm [%] vs. ligand concentration (M).

**Figure 3 cells-13-00033-f003:**
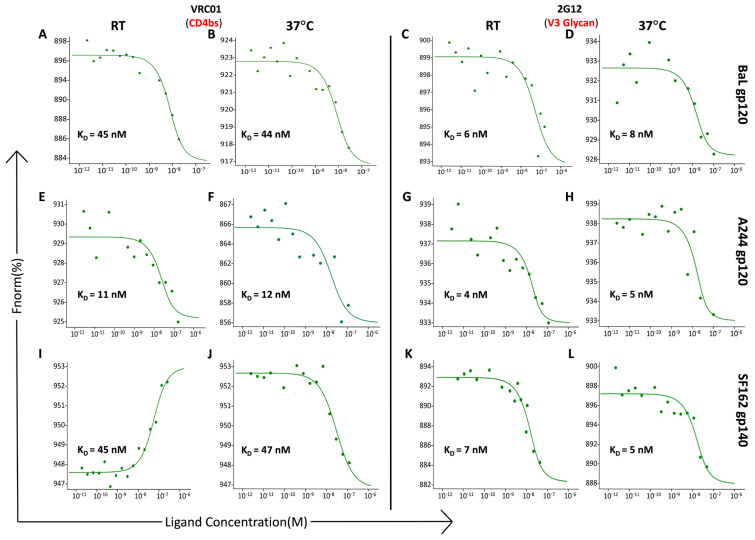
Impact of incubation temperature on the binding affinity of monoclonal antibodies (mAbs) to HIV-1 envelope proteins. Following a binding check, a constant concentration of the labeled mAbs (VRC01: 12.5 nM; 2G12: 15 nM) in DPBS buffer without Ca^2+^ and Mg^2+^, pH 7.2, and containing 0.01% Tween-20 was used for determining the binding affinity. Sixteen premium capillaries were loaded with VRC01 or 2G12 and serially diluted (1:1) HIV-1 Env proteins, BaL gp120, A244 gp120, and SF162 gp140 (100 nM, 150 nM and 75 nM, respectively), in a final volume of 10 µL. A Microscale Thermophoresis (MST) signal was then acquired using a NanoTemper Technologies Monolith NT.115 Pico instrument at an excitation power of 60% and an MST power of 60%. The signal was analyzed after the start of the IR laser, and the obtained data were fitted using MO. affinity analysis software provided by NanoTemper. The labeled mAbs and the serially diluted Env proteins were each incubated for 30 min at RT or 37 °C and then loaded into the capillaries for affinity measurement. MST traces and binding affinity data are as shown: (**A**,**B**,**E**,**F**,**I**,**J**) VRC01 and (**C**,**D**,**G**,**H**,**K**,**L**) 2G12 with BaL gp120 (**A**–**D**); A244 gp120 (**E**–**H**); and SF162 gp140 (**I**–**L**), respectively. Relatively similar binding affinities were obtained at the incubation temperatures tested. The graphs are presented as Fnorm [%] vs. ligand concentration (M).

**Figure 4 cells-13-00033-f004:**
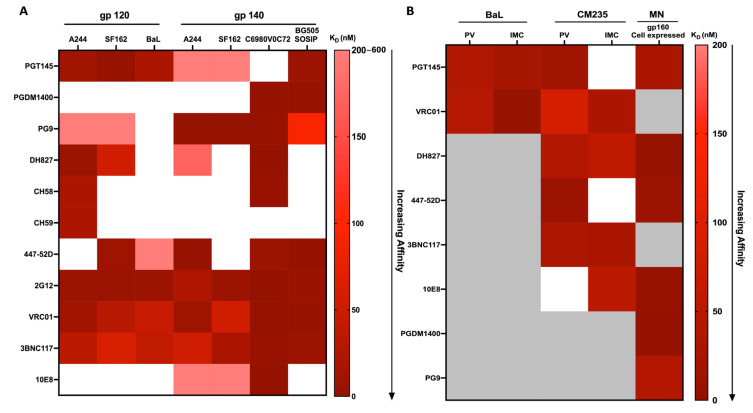
Heat maps depicting the binding affinities of monoclonal antibodies (mAbs) to recombinant HIV-1 envelope proteins, pseudoviruses (PVs), infectious molecular clones (IMCs), and cell- expressed gp160. (**A**) Binding affinities (K_D_) of mAbs to gp120 (A244, SF162, BaL) and gp140 (A244, SF162, C6980V0C72, BG505 SOSIP) proteins as determined by MST. (**B**) Binding affinities (K_D_) of mAbs to pseudoviruses (PVs), infectious molecular clones (IMCs), BaL and CM235, and cell surface-expressed gp160 (MN) protein. The color bar on the right side of each heat map exhibits the range of binding affinity (K_D_). Binding affinities (nM) are represented ranging from a lighter color (weaker affinity) to a darker color (stronger affinity). The absence of binding is represented by white, and grey indicates binding not determined. Heat maps were generated using GraphPad Prism software (version 10.0.1).

**Figure 5 cells-13-00033-f005:**
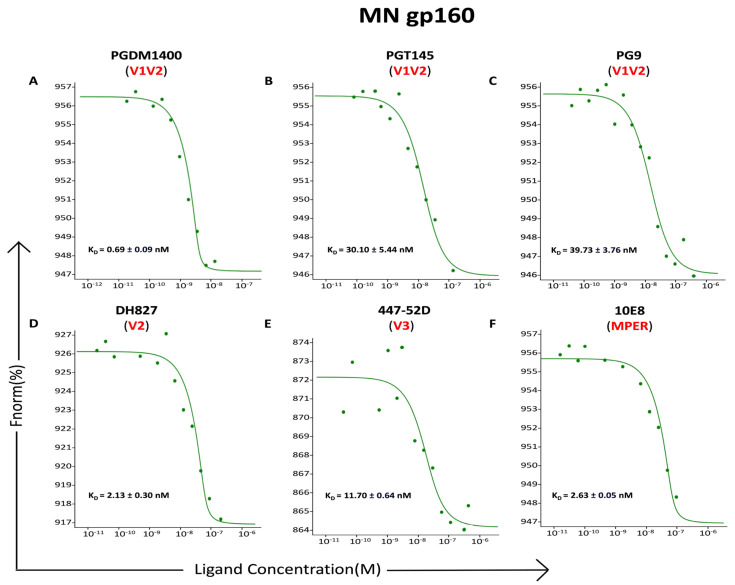
Microscale thermophoresis (MST) traces representing the binding affinities (K_D_) of monoclonal antibodies (mAbs) to a CEM cell line (CEM-NKR 5001A) expressing the MN gp160 HIV-1 Env protein on the cell surface. The labeled mAbs were used at the following final concentrations: (**A**) PGDM1400 (20 nM), (**B**) PGT145 (30 nM), (**C**) PG9 (30 nM), (**D**) DH827 (50 nM), (**E**) 447-52D (20 nM), and (**F**) 10E8 (50 nM). To determine the binding affinity, the capillaries were loaded with serially diluted (1:1) cells expressing the MN gp160 at a starting concentration of 5 × 10^6^ cells/mL in a final volume of 10 µL. Experiments were performed at room temperature for 30 min. The graphs are represented as Fnorm [%] vs. ligand concentration (M).

**Figure 6 cells-13-00033-f006:**
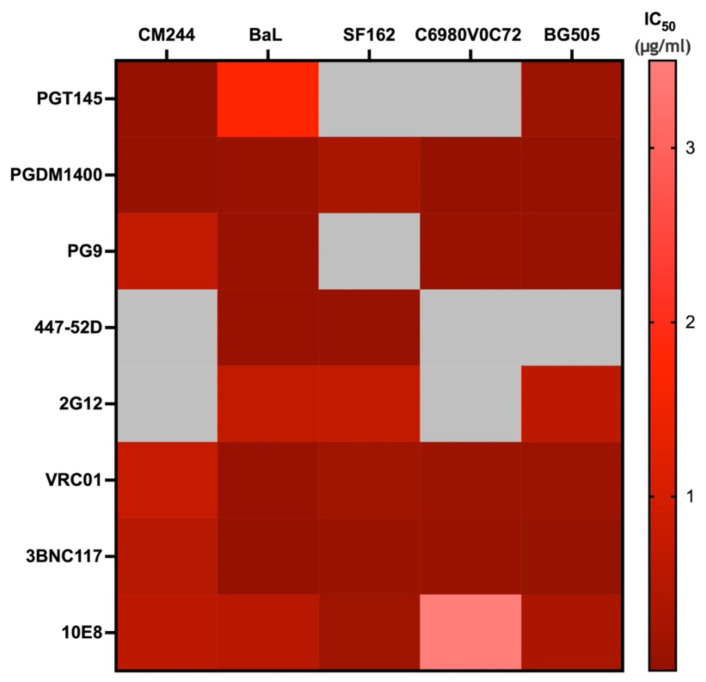
Heat map of neutralization values (IC_50_, μg/mL) obtained from the CATNAP database of monoclonal antibodies individually tested with different strains of HIV-1 (CM244, BaL, SF162, C6980V0C72, and BG505) for their neutralization potencies. The color bar on the right side indicates the range of neutralization potency IC_50_ (μg/mL). Darker colors indicate lower IC_50_ values (potent neutralization), while lighter colors indicate higher IC_50_ values (weaker neutralization). The grey color indicates neutralization not determined. Heat maps were generated using GraphPad Prism software (version 10.0.1).

**Figure 7 cells-13-00033-f007:**
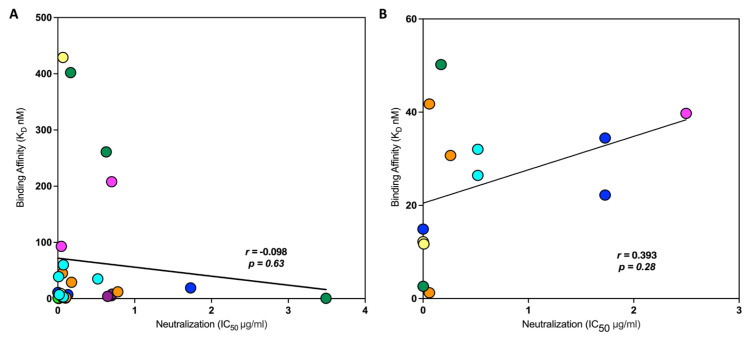
Pearson correlation analysis of neutralization (IC_50_, μg/mL) data vs. binding affinities (nM) of monoclonal antibodies (mAbs) with HIV-1 proteins, pseudoviruses (PVs), infectious molecular clones (IMCs), and MN gp160 (cell-expressed trimer protein), as determined by MST. X-axis represents neutralization values (IC_50_ μg/mL); Y-axis represents binding affinities (K_D_ nM). mAbs were differentiated in different colors (PGT145—blue, PGDM1400—fluorescent green, PG9—magenta, 447-52D—yellow, 2G12—purple, VRC01—orange, 3BNC117—cyan, and 10E8—dark green). There is no correlation, and it is not statistically significant. (**A**) Pearson correlation analysis of the binding affinities of the mAbs with the HIV-1 envelope monomer and trimer proteins (A244, BaL, SF162, C6980V0C72, and BG505) vs. neutralization data of the corresponding mAbs individually tested with similar strains of HIV-1 (CM244, BaL, SF162, C6980V0C72, and BG505). (**B**) Pearson correlation analysis of the binding affinities of mAbs with BaL and CM235 PVs, IMCs, and cell-expressed MN gp160 vs. neutralization data of the same strains of virus (BaL, CM235, and MN) with corresponding mAbs obtained from CATNAP. Correlation analyses were performed with GraphPad Prism v10.

**Table 1 cells-13-00033-t001:** Binding affinities of fluorescently labeled monoclonal antibodies (mAbs) to HIV-1 monomer and trimer proteins.

	gp120 Proteins			gp140 Proteins			
Monoclonal Antibodies (mAbs)	A244 (AE)	SF162 (B)	BaL (B)	A244 (AE)	SF162 (B)	C6980V0C72 (C)	BG505-SOSIP (A)
	K_D_ (nM) (Avg ± SD)						
PGT145	11 ± 1	3 ± 1	19 ± 8	287 ± 37	337 ± 8	NB	7 ± 0.4
PGDM1400	NB	NB	NB	NB	NB	0.3 ± 0.1	1 ± 0.3
PG9	208 ± 59	556 ± 21	NB	1 ± 0.2	1 ± 0.2	2 ± 0.3	93 ± 12
DH827	6 ± 0.02	54 ± 1	NB	171 ± 12	NB	0.5 ± 0.1	NB
CH58	20 ± 5	NB	NB	NB	NB	4 ± 2	NB
CH59	21 ± 3	NB	NB	NB	NB	NB	NB
447-52D	NB	9 ± 1	429 ± 63	3 ± 0.3	NB	6 ± 3	0.3 ± 0.03
2G12	6 ± 1	5 ± 1	8 ± 1	22 ± 2	6 ± 0.3	0.1 ± 0.01	4 ± 0.2
VRC01	12 ± 2	29 ± 3	45 ± 1	10 ± 0.2	55 ± 7	1 ± 0.3	2 ± 0.3
3BNC117	35 ± 4	60 ± 2	39 ± 3	53 ± 4	17 ± 4	3 ± 2	7 ± 1
10E8	NB	NB	NB	261 ± 5	402 ± 38	0.6 ± 0.03	NB

NB: no binding.

## Data Availability

All reagents will be made available upon request after the completion of a Materials Transfer Agreement. All data supporting the findings of this study are found within the paper and its Appendix A. Any additional information required to analyze the data reported in this paper is available from the corresponding author upon request.

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
