# Peer review of "Determination of Binding Affinity of Antibodies to HIV-1 Recombinant Envelope Glycoproteins, Pseudoviruses, Infectious Molecular Clones, and Cell-Expressed Trimeric gp160 Using Microscale Thermophoresis"

_cells, 2023, doi:10.3390/cells13010033_

Round 1

Reviewer 1 Report

Comments and Suggestions for Authors

This paper describes a new technique to measure antibody affinity in a way that does not require immobilization. This method has a potential to revolutionize such measurements as previously made by SPR or other methods that are costly and labor intensive and can not measure affnities with all kinds of antigens such as those involving antigens derived from membranes. The paper is quite interesting, but is rather hard going for the reader. This is partly because various different strains of envelope were used in different formats (gp120, gp140, PV, IMC, cells), some of which are tier 1 or the more vaccine relevant tier 2 strains. Moreover, the affinity data is presented as a series of graphs and tables coupled with a narrative that restates affinities and differences but does not “crunch” the data to show the patterns more clearly in one place – which would be difficult anyway because of the lack of use of the same strains throughout, which limits interpretations. The fact that so many different strains are included would be good if there was also a comprehensive analysis of 2 or 3 tier 2 strains. Also, the data really should be referenced to neutralization data for each mab virus pair – which is available publicly for pseudoviruses at least, - it would be interesting to know how IMC neutralization titers differ than PV neutralization titers and how that compares with affinities. This may well also be published (ref 59?). Affinity data could also be tabulated compared to previous studies data to verify affinities/consistency, which would be very valuable considering this is a new method. There is also a question of how PVs and IMCs were made (no methods are given) and how the concentration of envelope was determined in these cases, given that concentration should be for envelope, not total protein in PVs, IMC or cells, which would skew data. Overall, this method is an exciting opportunity to understand binding patterns and feed into vaccine development, but the patterns are far from clear as they are presented due to the many factors described above. 

Some comments are:

Line 32 abstract bNAbs and mab bound with higher affinity to native gp160 trimers on cells vs soluble? 

Line 81 A systematic characterization of the binding interaction of bNAbs with the various HIV-1 Env proteins has not been carried out” is perhaps a slight overstatement. There are papers out there using ELISA, flow cytometry, SPR, etc some of which are exhaustive (e.g., PMID: 24068931). However, I agree the picture is not fully clear if measurements use different assays for different formats. Using the same assay for different forms of Env would be valuable to create a clearer picture. Suggest instead to say a universal assay has not covered readings so there are some gaps in understanding how binding correlates with function and how forms of env differ. The fact it does not require immobilization, unlike ELISA, SPR, BLI, cell/VLP ELISA could be a benefit as they say, line 100.

Fig 1 text says PDB is 5FYJ, but the legend says 4TVP. Legend Text says dark green signifies MPER or unspecified gp41 (the lighter green seems to be the unspecified). 

Line 180: there is not a full explanation about what “the experiment was conducted to determine the binding affinity of the mAb” entails. I gather from later in the paper that it involves a repeat experiment in which the antigen is serially diluted. 

Fig S1 and S2:  What is the MAb and Env antigens used in this example data? What concentrqation of antigen (an excess, presumably).

Fig 2: the use of different Envs and antibodies for the temp and time experiments is not ideal. While the use of VRC01 with two different gp120s for time is OK, it is unclear why there is a switch to SF162 and 2G12 for the temperature test. Ideally a full test of both variables with two antibodies and two Envs would have been ideal. I do accept that the assay is likely fine, it is just that the too many variables and loose ends make this part not systematic enough to have full confidence – which is in fact an important gap created by similar inconsistent use of strains throughut the paper. The dots in panels A and B look to be absolutely identical (30 min and 60 min). That is not the case for the other data pairs. Is it possible that the wrong graph was shown for either part A or B?

Line 264: need to adjust the text: Of the trimers, PGT145 bound best to the BG505 SOSIP. This may be due to the improved “closed conformation”/folding (it binds slightly better to SF162 gp120, though). 

Why is the curve reversed VRC01 bottom middle Fig 3? I think the curves are great in this figure, but they could be made supplemental as they are perhaps not really the best way to compare binding data between antibodies and Envs. I further suggest that Tables 1 and 2 and possibly other data (Fig 4 and 5) is formatted into a single heat map figure so that the patterns of higher or lower affinity are more obvious and accessible to the reader. Graphs could be used to compare affinities between PV verus protein, PV vs IMC and possibly neutralization versus these affinities. It would in general good to see two or more strains examined in gp120, gp140, PV and IMC/cell formats, to dissect these questions, preferably using tier 2 strains (i.e., not. MN, SF162, BaL and other V3-sensitive tier 1 strains). In this way, the affinity data would acquire greater meaning and impact as we can cross-compare the effects of the different formats. At present, the data is very fragmented and difficult to “process”. Showing the graphs and multiple tables has a diluting effect on the impact of the findings and also makes the narrative hard going. The text saying x is y fold lower affinity to a than b as well as rewriting epitopes and kD’s in the text when they are already in the tables/graphs -- is laborious and distracting for the reader, whereas a centralized heat map would enable the reader to see if for example a given antibody binds better to gp120 than a trimer and the narrative could say that without getting bogged down by fold differences which can be inferred from the heat map and sparingly mentioned in the text for the most significant pieces of data. In short, the text merely restates the data, but doesn’t interpret it or go far enough to identify any patterns. Also, the heat map should include neutralization data for MAb-virus pairs (available in CATNAP and various published papers) so a correlation can be made between the two (and possibly could be plotted to check for correlations). 

It is good to see the modestly neutralizing V3 MAb 447-52D binding BG505 so well. V3 MAb binding is scarcely acknowledged by SOSIP researchers, so it is good to see that transparency. 

Line 291: I don’t think epitopes of mabs need to be mentioned again, as this is covered in fig 1. Perhaps the best approach is to deal with all V2 mAbs under a subheading (PGT145, PGDM, PG9 etc), then move to V3 subheading etc, for easy reading. Similarly, subtypes in Tables 1 and 2 could be abbreviated as AE, B, C etc so there is more focus on data. Same for the +/- on the affinities. It would be good to say what the range of standard deviation from the curve is (e.g., a max of 10% each way), so the kDs are emphasized more for clarity. 

Line 293:”we observed a similar result with PG9” (similar to what?)

Line 301: regarding the non-neutralizing mAbs, it would be good to recap the immunogens used and say if they bind to the gp120 (A244 gp120 is one of them) or other Envs within the immunogens. Is binding strain specific as might be expected? 

Line 310: it would be good to sum up the V2 mAbs in a sentence, perhaps how their binding profiles differ, if it tracks with neutralizing activity etc? There need to be sentences to summarize the complex behaviors for the more casual readers. 

Fig. 3, Tables 1 and 2: Were the oligomeric states of the different gp140s checked, or are they known? While SOSIP is probably a trimer, the other gp140s may produce various forms, including aggregates, so it is unclear what forms of gp140 the binding measures (possibly monomer?). 

It would be good to speculate about why 10e8 neutralize BG505 (CATNAP database) but does not bound to trimer gp140 by MST. Is the epitope intact on BG505 (not truncated? The epitope is near the C-terminus of the MPER which may be more truncated in SOSIP than the others which enabled the designers to obtain a crystal structure). In addition, comments could be made on why certain mAbs do not bind to gp120 or gp140 (e.g., 447-52D does not bind to A244 gp120, but binds to A244 gp140). 

Fig. 4: Similarly, what is the oligomeric state/conformation of Envs on PV and IMC? While some envelope is native, it is well-documented that PVs incorporate other forms of Env. IMCs may be somewhat cleaner, but that also depends on the producer cell. (e.g., 293T cells versus PBMCs). Suggest to run them on gels to address the difference between PV and IMC as mentioned in the discussion. Was a comparison made between MST affinity vs neutralization of PV/IMC by the same mAbs? 

Line 367 PGT145 binding to the IMC is stronger not the other way around as stated. 

Line 368 “In contrast,” could be replaced by “Similarly, as the IMC is bound more strongly by both MAbs, although the increase is more notable for VRC01. 

The methods of manufacture of PV and IMC should be included in the methods section, given that this is discussed in lines 493-498. Also, how are particles concentrated, lysed, lysis presumably needs more than 0.01% tween initially, but may be OK if lysed then diluted. Line 496 “The gp41 endodomains were also different.”- It is not clear what this means and no explanation is provided as there are no methods given for the production of IMC and PV. The spike density of envelope on particles difference (line 498) should not impact the data because particles are lysed before the assay? The comment line 502 about envelope incorporation differences could be verified with the test samples by running them in gels. 

How are Env concentrations determined for the PV, IMC and cells expressing Env? This is complicated by their format which precludes e.g., nanodrop. For example, are the antigens serially diluted in a protein gel/Western blot/dot blot and compared to a cognate strain of gp120 also serially diluted to estimate the gp120 concentration in a given prep. This would seem to be important for the affinity to be determined accurately. 

For Membrane forms of Env, what happens to the hydrophobic transmembrane domain? In detergent? Can they mimic the membrane? MPER antibodies are affected by membrane as they partially bind lipid and/or their epitopes are partially occluded by lipid. 

Line 454: The discussion should, where possible (when the same strain/envelope format is used elsewhere), compare MST data to other methods published by other groups. How is data similar or different from flow cytometry, and other published methods such as SPR, BLI, ELISA? It would be good to tabulate the previously published data and/or compare it maybe in a scatterplot to see if patterns are consistent by this method at least for BG505 SOSIP as this is likely the most published with various mAbs by different methods and neutralization data is also known. One example of previous BG505 SOSIP binding and neutralization data is in PMID: 24068931, but there are many other papers. 

Lines 164-165. Check line spacing.

Line 411: No binding was observed with the parental cell line. This data should be included in the supplementary section.

Line 504: the expression of MN on cell surfaces should not be any better as a representative of native conformation than PV or IMC?, it’s just another on membrane format that is cells not particles. A problem with the MN strain and tier 1 isolates used in these formats is that gp120 shedding might impact data. This adds to the other problems mentioned above and by the authors that envelope on surfaces may be not completely processed and there could be some different forms presented that have different affinities. If there is more than 1 form of envelope in a sample,, with different affinities, its not clear how that might impact the data. Would affinities reflect binding to both forms, or the strongest of the two or the most highly expressed? 

Reviewer 2 Report

Comments and Suggestions for Authors

Comments to Manuscript ID: cells-2674609

The manuscript uses microscale thermophoresis (MST) to analyze the binding affinity of broadly neutralizing antibodies to HIV-1 envelope (Env) glycoprotein made in the format of recombinant soluble protein, presented on the surface of transfected cells, or on the surface of pseudovirions (PV) or infectious molecular clone (IMC). This is the first time MST is being used in HIV-1 study and potentially can be used in parallel with other established methods such as surface plasmon resonance (SPR) or bio-layer interferometry octet (BLI).

Overall, the manuscript is well written, albeit lacking details in the Materials and Methods. In addition, the discussion was not in-depth (see below comments).

General comments:

1.    Line 58: Reference “3” is inappropriate. Should use the appropriate reference that report these co-receptors.

2.    Line 69-72: Fusion peptide is one of the more recent bNAb epitope.

3.    Avoid starting sentence with “Also”. Lines 84, 462.

4.    Lines 164-165. Check the line spacing.

5.    Recommend including a short narrative on how PV and IMC is made, given that this is discussed in Lines 493-498. 

6.    Y-axis of all Fnorm graphs should be standardized.

7.    Be specific about “ligand concentration” in all graphs. Is it referring to antibody?

8.    Figure 1: Recommend using another color for gp41, to differentiate from MPER epitope. In addition, remove the “bracket” for V1V2.

9.    This is personal preference. Table 2 and 3 can be consolidated into one large table and the Kd values can be color coded for easy data interpretation. For eg, A244 gp120 and A244 gp140 can be put side-by-side for clarity. In addition, the “subtype” word can be removed from the table to save space.

10. Line 411: No binding was observed with the parental cell line. This data should be included in the supplementary section.

11. Table 2: Unbold PGT145 and some of the Kd values.

Questions to authors:

1.    Figure S1: What is the antibody used? Do all the labeled mAbs behave similarly?

2.    Figure S2: What is the antigen used? Is there antigen that do not bind to the mAbs as control? For eg, non-HIV-1 proteins. This is important to ensure that the assay is specific for HIV-1 Env glycoprotein.

3.    Line 176: What is the rationale in using different concentration for different mAbs? Do you think this will affect the antibody-antigen stoichiometry/binding affinity? In addition, in all MST assays, what is the Env glycoprotein concentration used? Will the Env glycoprotein concentration affect the outcome of MST assay? Lastly, certain mAb can bind to 2 Env at the same time. Does this assay able to differentiate/identify one NAb-one Env complex or one NAb-2 Env complex? This information will be useful in understanding the stoichiometry of NAb.

4.    Line 188: Justify 1.5 second to record the signal.

5.    What are the reasons to use Tier 1 Env (BaL, SF162, MN)? Do you think the binding affinity will be different if used Tier 2 and/or 3 Env? Suggest including Tier 2 and/or 3 Env in this assay to demonstrate the robustness of the assay.

6.    Figure 2A-D, what temperature was this experiment conducted?

7.    Figure 2E-F, how long was the incubation? 30 mins or 60 mins? This information can be included in the figure legend. Will there be a temperature effect if this was done on gp120? Lastly, how many times was this experiment repeated? It would be good to indicate the standard deviation of the Kd values.

8.    Figure 3: Why is BG505 SOSIP vs VRC01 graph different from the rest?

9.    Was the oligomeric state of the Env glycoprotein checked on SDS-PAGE/BN-PAGE prior to MST experiments? This information is useful to ascertain that gp140 are trimer.

10. Were the PV and IMC Env oligomeric state being check on gels? This would address the difference between PV and IMC as mentioned in the discussion (Lines 493-499).

11.  Comment on why 10e8 can neutralize BG505 (Catnap) but not bound to trimer gp140 by MST. In addition, lacking comments on why certain mAbs do not bind to gp120 or gp140 (eg not limited to: 447-52D do not bind to A244 gp120, but binds to A244 gp140).

12.  Do you think MST can be used to measure CD4i binding in the presence of soluble CD4?

13. Figure 5: Why are the line graphs differing from Figure 4? What temperature was used for this experiment and what is the incubation time? In addition, how do you measure PV and IMC concentration (Line 398). Indicate this in the methods.

14. MN gp160 on cell surface cannot be compared side-by-side with SF162 gp140 soluble as they are 2 degrees separation (different strain, different type of protein). It would be good to make cell surface expressed SF162 for direct comparison.

15. The discussion did not compare the MST result to other methods published by other groups. An important question to authors is that how this method differs from flow cytometry whereby both methods use labeled antibody and flow cytometry has been used to measure binding to cell surface Env. In addition, how does MST data stack against other published methods such as SPR, BLI, ELISA, virus capture. One thing that was not mentioned is the cost of MST in relation to other methods. This is crucial given that MST is being highlighted as “First” to be used for such analysis.

16.  Was there comparison done between MST binding affinity vs neutralization of PV/IMC by the same antibodies? This would be useful to investigate, given that the future step is to evaluate polyclonal sera for potential isolation of bNAb(s). This was mentioned in Lines 527-529. However, given that MST is constantly highlighted in the manuscript as being first, this comparison can be done to showcase the importance of MST.

17.  Recommend showing a graphical illustration (as summary) of the epitope exposure on different Env formats (soluble, Env on cells, Env on PV/IMC) determined by MST vs other methods. This can highlight the importance of MST in analyzing binding affinity of mAbs/nAbs to Env glycoprotein.

Comments on the Quality of English Language

No issues.

Reviewer 3 Report

Comments and Suggestions for Authors

In this manuscript, the authors described the application of a biophysical method (microscale thermophoresis, MST) to measure the affinities of antibodies binding to HIV-1 envelope (Env) in the forms of recombinant protein (monomer or trimer), or on the surface of pseudoviruses (PVs), infectious molecular clones (IMCs), or cells. As the authors pointed out, a systematic characterization of the binding interaction of bNAbs with the various HIV-1 Env proteins has not been done. The work is systematic, thorough and well described. This manuscript will be a useful reference for the field to facilitate studies such as vaccine design (to monitor binding affinity of elicited antibodies) and antibody engineering (to enhance affinity of antibodies binding to various intact viral particles).

Comments:

1.                   The authors mentioned on page 2 that “Also, from a therapeutic standpoint, it would aid in formulating a better cocktail of bNAbs and/or mAbs that would be more effective globally.”. This is a strong statement which could use some supporting evidence. Translating bNAbs’ in vitro anti-viral activity to in vivo efficacy is already not straightforward (Burton Nat Rev Immunol. 2023 Apr 17:1-15), predicting in vitro potency from binding affinity is one extra step. If the authors could list some reference supporting the correlation between bNAbs’ affinity and anti-viral potency that will be helpful.

2.                   Related to the 1st point, in Figure 5 where the authors observed binding of certain bNAb’s to only pseudovirus (PV) but not infectious molecule clone (IMC) (or vice versa), can the authors test if the bNAb for activity against both the PV and IMC? If the antibodies are still active in both anti-viral assays, does that mean the binding assay and anti-viral assay have a different level of sensitivity?

3.                   Not quite sure if it’s the submission or something happened when it downloaded, but the line numbers are not accurate.

Round 2

Reviewer 1 Report

Comments and Suggestions for Authors

Basu et al, re review. 

This is a great revision, much improved.  

The Fig. 4 heat map is effective. For example, we can infer that some bnAbs higher affinity to IMCs than pseudoviruses. However, generalizations are difficult, complex.

Some small points are: 

Fig. 7: what are the mAbs in each symbol? A legend would be good . In part B, there seems to be an outlier? This might affect the correlation? 

Minor: in abstract: “Interesting differences” - differences need to be described more clearly than “interesting”. Do the differences make any sense. Do they have a pattern or is it that the epitope exposure/occlusion profiles differ, suggesting conformational differences? 

Line 75 edit for clarity: …mAbs show variable affinities”

Line 78 edits: “the various platforms” I think means soluble monomers, trimers and membrane expressed forms of Env. For me, the benefit here is that the current method is not limited to soluble proteins like SPR/BLI where immobilization is needed (mentioned in the next paragraph- I think it would be good to re-organize the text so the sentences on lines 87-92 are moved to the beginning of a new paragraph that would start with the text of lines 87-92 and continue with the red text at line 76. Membrane forms of Env more likely to be done by flow cytometry, cell ELISA, VLP ELISA. The use of different platforms makes it difficult to compare mAb affinities to membrane or soluble forms of Env. However, the current methods can analyze these proteins “under one roof”. 

Line 82 new text. I don’t understand why this is added. If this is to bring avidity into the conversation, it needs a lead in sentence to explain why. Also, does the current method measure avidity? 

Line 108 “and recombinant proteins” text could be deleted (the monomer and trimer are recombinant is said previously in this sentence). 

Line 132: “fan blade” does this imply an open or closed trimer (closed like SOSIP usually is)? It is useful to explain this as these conformations affect antigenic properties. In general, it seems like the envelope samples come from a variety of sources. 

Line 153. The methods for particles are good. Thank you for providing them. That said, the paper would be a bit more powerful if a full set of samples (gp120, soluble trimer, PV, IMC etc) were all made in the same strain. As it is, samples were used based on availability and in some cases previous use in vaccines. On the other hand, that would take some time to generate which may not satisfy the urgency of this to get the message about this new method out first, which is understandable. There is sufficient cross-comparisons of protein types within some different subtypes to satisfy this point and, importantly, to showcase the method.

Line 212 remove ligand, this is actually antigen anyway.!

Comments on the Quality of English Language

The intro could still be improved slightly, but it is much better this time

Reviewer 2 Report

Comments and Suggestions for Authors

Thank you for addressing all the comments and thank you for the great work.

Author Response

We thank the reviewer for the appreciating our work.